# Survival of Myocardial Infarction Patients with Diabetes Mellitus at the Invasive Era (Results from the Városmajor Myocardial Infarction Registry)

**DOI:** 10.3390/jcm12030917

**Published:** 2023-01-24

**Authors:** Réka Skoda, Attila Nemes, György Bárczi, Hajnalka Vágó, Zoltán Ruzsa, István F. Édes, Attila Oláh, Annamária Kosztin, Elek Dinya, Béla Merkely, Dávid Becker

**Affiliations:** 1Heart and Vascular Center, Semmelweis University, 1085 Budapest, Hungary; 2Department of Medicine, Albert Szent-Györgyi Medical School, University of Szeged, 6720 Szeged, Hungary

**Keywords:** diabetes mellitus, acute myocardial infarction, prognosis

## Abstract

Due to the lifelong nature of diabetes mellitus (DM), it has been demonstrated to have significant effects on patients’ morbidity and mortality. The present study aimed to assess the effects of DM on the clinical outcome and survival in patients who underwent percutaneous coronary intervention (PCI) due to myocardial infarction (MI) and to examine the relationship of DM to the type of the MI and to left ventricular (LV) and renal functions. A total of 12,270 patients with ST-elevation MI (STEMI) or non-ST-elevation MI (NSTEMI) were revascularized at our Institution between 2005 and 2013. In this pool of patients, 4388 subjects had DM, while 7018 cases had no DM. In both STEMI and NSTEMI, the 30-day and 1-year survival were worse in diabetic patients as compared to non-diabetic cases. In the patients with DM, NSTEMI showed worse prognosis within 1-year than STEMI similarly to non-diabetic subjects. Regarding survival, the presence of DM seemed to be more important than the type of MI. Regardless of the presence of DM, reduced LV function was a maleficent prognostic sign and DM significantly reduced the prognosis both in case of reduced and normal LV function. Survival is primarily affected by LV function, rather than DM. Worse renal function is associated with worse 30-day and 1-year survival in both cases with and without DM. Considering different renal functions, the presence of DM worsens both short- and long-term survival. Survival is primarily affected by renal function, rather than DM. The results from a high-volume PCI center confirm significant the negative prognostic impact of DM on survival in MI patients. DM is a more important prognostic factor than the type of the MI. However, survival is primarily affected by LV and renal functions, rather than DM. These results could highlight our attention on the importance of recent DM treatment with new drugs including SGLT-2 inhibitors and GLP-1 antagonists with beneficial effects on survival.

## 1. Introduction

Diabetes mellitus (DM) is a significant risk factor for cardiovascular disorders, as the abnormal metabolic state increases the risk for atherosclerosis and consecutive vascular occlusive disorders [1]. Due to the lifelong nature of DM, it has been demonstrated to have significant effects on patients’ morbidity and mortality [2]. The presence of DM is associated with poor prognosis in heart failure (HF) and following myocardial infarction (MI) [3,4,5]. 

Left ventricular (LV) ejection fraction (EF) as a surrogate marker of LV systolic function is known to be a strong predictor of cardiovascular outcome and survival even in HF or following MI [6]. Similarly, renal function, as manifested by the glomerular filtration rate (GFR), has a graded inverse association with the burden of cardiovascular comorbidities and long-term adverse events [7]. Limited information is available, however, on the relationship of DM to the type of MI [ST-elevation (STEMI) vs. non-ST-elevation MI (NSTEMI)], to LV-EF, and to GFR, as well as on the clinical outcome in patients following percutaneous coronary interventions (PCIs) due to acute MI. Therefore, the present study aimed to assess the effects of DM on clinical outcome and survival in patients who underwent PCI due to MI in a high-volume PCI center within the framework of MI management for Central Hungary and to examine the relationship of DM to the type of MI and to left ventricular (LV) and renal functions.

## 2. Materials and Methods

### 2.1. Patient Population

A total of 12,270 patients with STEMI or NSTEMI have been revascularized at our Institution between 2005 and 2013. In this pool of patients, 4388 subjects had DM, while 7018 cases had no DM. These patients were enrolled in a registry named Városmajor Myocardial Infarction Registry (VMAJOR-MI Registry), in which all the available demographic and clinical patient data have been summarized. During the follow-up, all the patients were tried to be followed up by phone, mail, or other available way. The follow-up data regarding the primary outcome were confirmed from hospital recordings or autopsy reports, where available. The short-term 30-day and long-term 1-year survival data were examined in patients admitted with NSTEMI (n = 6840) and STEMI (n = 5430).

For the diagnosis of STEMI, a clinical presentation typical of ongoing myocardial ischaemia, ST-segment elevation in at least 2 consecutive leads on the 12-lead electrocardiogram (ECG), and subsequent confirmation by elevation of necrosis markers were required [3,4]. For the diagnosis of NSTEMI, only necrosis marker elevation with typical symptoms were required. The definition of DM was based on the American Diabetes Association [8] and World Health Organization criteria [9]. Transthoracic echocardiography was performed in all the cases in accordance with applicable guidelines [2]. The LV function was represented by LV-EF. The laboratory examinations included the evaluation of renal function using GFR in all the subjects. The severity of renal dysfunction was established according to recent guidelines [10].

Only the patients with primary revascularization defined as percutaneous coronary intervention (PCI) were included in the study [6,8]. Coronary stenosis or occlusion was evaluated from multiplane projections and a luminal diameter reduction of >50% was considered significant. The PCI was performed according to the contemporary guidelines [11]. 

The study protocol conformed to the ethical guidelines of the 1975 Declaration of Helsinki and was approved in advance by the locally appointed ethics committee (30088-2/2014/EKU). The primary outcome of the study was all-cause mortality. The National Health Care Institute provided the accurate details on the above endpoint with occurrence dates.

### 2.2. Statistical Analysis 

The results are expressed as mean ± standard deviation of mean (S.D.) and sample size (n) for each treatment group with normal distribution. The normal distribution of data was checked by applying the Shapiro–Wilk’s test. When the non-normally distributed data were analyzed, the homogeneity of variances was assessed using a Levene’s test. The means were compared using a Student’s t-test in case of normal distribution and (ii) a Mann–Whitney-Wilcoxon test was used for datasets that were not normally distributed. A separate analysis of variance (ANOVAs) with a Tukey’s correction for multiple comparisons was applied where appropriate. A Pearson chi-square test (*χ*^2^) or Fisher’s exact test was applied in the case of categorical data. The Kaplan–Meier product-limit method of survival analysis was used to summarize the follow-up. The differences in survival rates between groups were tested using a long-rank test. The analysis was two-sided, with a level of significance of *α* = 0.05. All the statistical analyses were performed using the SAS 9.4 (SAS Institute Inc., Cary, NC, USA) software package. For offline data analysis and graph creation, a commercial software package was used (Microsoft Excel 2016).

## 3. Results

### 3.1. Demographic and Clinical Data 

The demographic and clinical data with significant differences between patients with and without DM are presented in Table 1. The type of MI (NSTEMI vs. STEMI), LV-EF, and renal function differed significantly between the patients with vs. without DM (Table 2). 

### 3.2. Diabetes Mellitus and the Type of Myocardial Infarction 

In both STEMI and NSTEMI, the 30-day (*p* = 0.0022 and *p* = 0.0029, respectively) and 1-year survival (*p* < 0.0001 and *p* < 0.0001, respectively) were worse in diabetic patients as compared to non-diabetic cases. In the patients with DM, NSTEMI shows worse prognosis within 1 year than STEMI (*p* = 0.0097); the results are similar in non-diabetic subjects (*p* = 0.0175). Similar findings were seen at the 30-day follow-up (*p* = 0.3582 and *p* = 0.4157). The STEMI patients without DM had the best survival chances and the NSTEMI patients without DM had similar chances to diabetic STEMI patients, but the diabetic NSTEMI patients had the worst prognosis (Table 3). Regarding survival, the presence of DM seemed to be more important than the type of MI (Figure 1).

### 3.3. Diabetes Mellitus and Left Ventricular Function 

Reduced LV function is a maleficent 30-day and 1-year prognostic sign regardless of the presence (*p* < 0.0001 and *p* < 0.0001, respectively) or absence (*p* = 0.0007 and *p* < 0.0001, respectively) of DM. DM significantly reduce the 30-day and 1-year prognosis both in the case of reduced (*p* = 0.0007 and *p* = 0.0004, respectively) or normal (*p* = 0.02 and *p* < 0.0001, respectively) LV function. It is not surprising that non-diabetic patients with acute coronary syndrome (ACS) and with normal EF had the best prognosis, as presented in Table 3, as diabetic patients with reduced EF had the worst survival chances. The analysis of survival showed that it is primarily affected by LV function, rather than DM (Figure 2, Table 3).

### 3.4. Diabetes Mellitus and Renal Function

Worsening renal function is associated with worse 30-day (GFR > 60 mL/min vs. GFR 30–60 mL/min: *p* < 0.0001 and GFR 30–60 mL/min vs. GFR < 30 mL/min: *p* = 0.0018, respectively) and 1-year (GFR > 60 mL/min vs. GFR 30–60 mL/min: *p* < 0.0001 and GFR 30–60 mL/min vs. GFR < 30 mL/min: *p* = 0.0001, respectively) survival in the patients with DM. A similar relationship could be demonstrated in the non-diabetic patients (30-day; GFR > 60 mL/min vs. GFR 30–60 mL/min: *p* < 0.0031 and GFR 30–60 mL/min vs. GFR < 30 mL/min: *p* = 0.0329, respectively) and 1-year survival (GFR > 60 mL/min vs. GFR 30–60 mL/min: *p* = 0.0001 and GFR 30–60 mL/min vs. GFR < 30 mL/min: *p* = 0.0001, respectively). The presence of DM worsens both short- (GFR > 60 mL/min: *p* = 0.0280, GFR 30–60 mL/min: *p* = 0.0019 and GFR < 30 mL/min: *p* = 0.0157) and long-term (GFR > 60 mL/min: *p* = 0.0004, GFR 30–60 mL/min: *p* < 0.0001 and GFR < 30 mL/min: *p* = 0.0032) survival. The analysis of survival showed that it is primarily affected by renal function, rather than DM (Figure 3, Table 3).

## 4. Discussion

DM is a group of physiological dysfunctions due to hyperglycaemia resulting directly from insulin resistance or inadequate insulin secretion. The individuals with DM are at high risk for both micro- and macrovascular complications including nephropathy, neuropathy, and occlusive cardiovascular comorbidities [11]. LV function is highly dependent on these functions and is usually represented by LV-EF, which has a known significant prognostic impact in several clinical scenarios [6]. Diabetic nephropathy is a clinical symptom with progressive decline in the GFR, which has a prognostic significance as well [7]. 

The presence of DM per se adversely affects the 5-year survival and risk of hospitalization in patients with acute and chronic HF [5]. No significant differences were found in adverse events between new and previously diagnosed DM patients. DM and newly diagnosed DM were independent predictors of 3-year mortality and 3-year major adverse cardiac events. The patients with newly diagnosed DM had a similarly poor prognosis after primary PCI in STEMI as those with previously established DM [4]. While T2DM increased the long-term mortality rate of elderly MI patients, the younger patients with both MI and T2DM had more complications in the early post-MI period compared with the patients of the same age group without T2DM. The younger patients with MI and T2DM did not show any statistically significant differences in the long-term outcome [3]. 

The present study aimed to assess the effects of DM on outcome and survival in the patients who underwent PCI due to MI and the relationship of DM with the type of MI, LV, and renal functions. Our results confirmed that survival was worse in DM patients when compared to non-diabetic cases regardless of the type of MI. However, the DM patients with NSTEMI had worse prognosis than the STEMI subjects suggesting that DM is a more important factor than the type of MI for prognosis. Regardless of the presence of DM, reduced LV function was found to be a maleficent prognostic sign and survival was primarily affected by LV function, rather than DM. Moreover, regardless of the presence of DM, worse renal function was associated with worse survival. Similar to LV-EF, survival was primarily affected by renal function, rather than DM. 

Together with optimal treatment strategies to improve renal function, the results of the present study could highlight our attention on the importance of using novel antidiabetic regimes with confirmed beneficial prognostic impacts on survival including sodium-glucose cotransporter-2 (SGLT-2) inhibitors and glucagon-like peptide-1 (GLP-1) receptor agonists in diabetic patients [12]. While SGLT-2 inhibitors have a beneficial effect on the prevention and progression of heart failure based on data from trials such as EMPAREG-OUTCOME, CANVAS, or DECLARE-TIMI 58, GLP-1 receptor agonists were confirmed to have preventive effects on all-cause hospitalizations based on trials such as REWIND [13]. 

There is a need for an appropriate diagnostic tool for the early diagnosis of acute MI. Circulating miR-19a was found to have a prognostic value as a promising molecular target with acceptable sensitivity and specificity for the early diagnosis and prognosis of acute MI [14]. Moreover, the potential of microRNA-1 and microRNA-221–3p as informative biomarkers positively correlating with coronary artery stenosis in MI was found [15]. Moreover, the microRNA reducing effects of metformin can provide a potential therapeutic strategy for patients with type 2 diabetes by reducing the MI risk [16].

There is a substantial difference in the pathomechanism of STEMI and NSTEMI with less total acute occlusion and more extensive vascular disease with multiple narrowing in the latter [17,18]. Advanced DM accelerates vascular smooth muscle cell dysfunction, which contributes to the development of vasculopathy [19]. Due to the effects of DM on vascular atherosclerosis, worse prognosis was expected and found in NSTEMI [20]. However, according to the findings of the present study, the presence of DM seems to be more important than the type of MI demonstrating the prognostic importance of DM in both forms of MI for survival.

## 5. Limitations

The present study was a retrospective observational study. Further prospective studies are warranted to confirm the results. In several cases, the cause of death was not known, therefore only all-cause mortality data are presented without any detailed information. Unfortunately, there was no information on the type of DM in any of the cases, therefore the term ‘DM’ was used throughout the manuscript.

## 6. Conclusions

The results from a high-volume PCI center confirm the significant negative prognostic impact of DM on survival in MI patients. DM is a more important factor than the type of MI for prognosis. However, survival is primarily affected by LV and renal functions, rather than DM. These results could highlight our attention on the importance of recent DM treatment with new drugs including SGLT-2 inhibitors and GLP-1 antagonists with beneficial effects on survival.

## Figures and Tables

**Figure 1 jcm-12-00917-f001:**
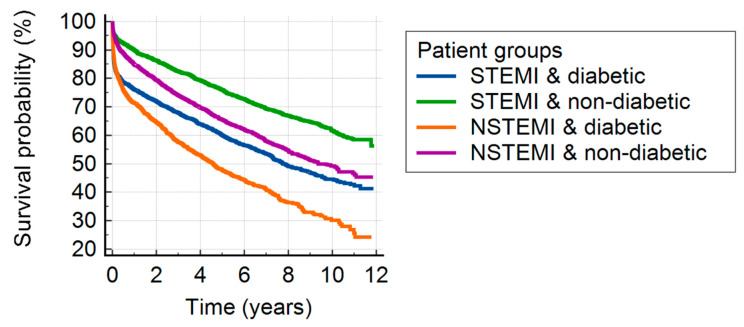
Kaplan–Meyer curves illustrating survival in diabetic versus non-diabetic patients with ST-elevation versus non-ST elevation myocardial infarction. Abbreviations: STEMI = ST-segment elevation myocardial infarction, and NSTEMI = non-ST-segment elevation myocardial infarction.

**Figure 2 jcm-12-00917-f002:**
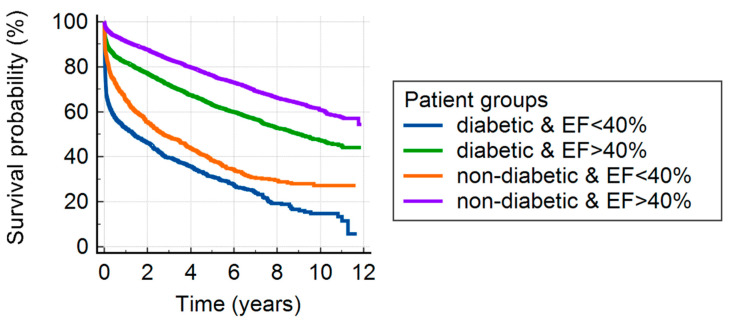
Kaplan–Meyer curves illustrating survival in diabetic versus non-diabetic patients with reduced versus normal left ventricular function. Abbreviations: EF = left ventricular ejection fraction.

**Figure 3 jcm-12-00917-f003:**
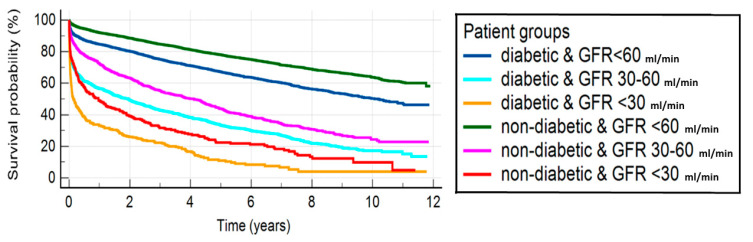
Kaplan–Meyer curves illustrating survival in diabetic versus non-diabetic patients with worsening renal function. Abbreviations: GFR = glomerular filtration rate.

**Table 1 jcm-12-00917-t001:** Demographic and clinical data of patients.

	Patients with DM	Patients without DM	*p*
mean age (years)	68.0 ± 12.7	65.0 ± 13.3	<0.0001
male (%)	57.7 (2533/4388)	65.32 (4585/7018)	<0.0001
mean BMI (kg/m^2^)	28.6 ± 5.4	27.4 ± 5.4	<0.0001
mean survival (days)	1560 ± 1201	1971 ± 1180	<0.0001
30-day survival (%)	13.5 (586/4330)	4.3 (299/6987)	<0.0001
1-year survival (%)	25.7 (1111/4330)	12.4 (869/6987)	<0.0001
NSTEMI (%)	45.7 (1916/4195)	55.6 (3701/6662)	<0.0001
STEMI (%)	54.3 (2279/4195)	44.5 (2961/6662)	<0.0001
ventricular fibrillation (%)	6.9 (300/4334)	3.2 (221/6940)	0.06
cardiogenic shock (%)	10.2 (442/4334)	2.6 (179/6942)	0.002
heart failure (%)	26.8 (1162/4334)	12.9 (892/6942)	<0.0001
mean LV-EF (%)	46.0 ± 13.8	50.2 ± 11.3	<0.0001
mean cholesterol (mmol/L)	4.62 ± 1.39	4.68 ± 1.35	0.0429
mean LDL-cholesterol (mmol/L)	2.98 ± 1.31	3.03 ± 1.3	0.0899
mean triglyceride (mmol/L)	1.57 ± 1.45	1.4 ± 1.07	<0.0001
mean GFR (mL/min)	72.88 ± 30.36	83.7 ± 31.15	<0.0001
mean creatinine (um/L)	104 ± 69.3	92.5 ± 66.9	<0.0001
mean peak Troponin (ng/L)	2478 ± 4560	1613 ± 3010	<0.0001

Abbreviations. DM = diabetes mellitus, GFR = glomerular filtration rate, LDL = low-density lipoprotein, LV-EF = left ventricular ejection fraction, NSTEMI = non-ST-elevation myocardial infarction, STEMI = ST-elevation myocardial infarction, and BMI = body mass index.

**Table 2 jcm-12-00917-t002:** Survival of patients with versus without diabetes mellitus.

Mean Survival (Days)			
	Patients with DM	Patients without DM	*p*
STEMI (n)	1770	2206	<0.0001
NSTEMI (n)	1341	1773	<0.0001
LV-EF < 40%	970	1203	<0.0001
LV-EF > 40%	1784	2085	<0.0001
GFR > 60 mL/min	1863	2152	<0.0001
GFR 30-60 mL/min	1080	1351	<0.0001
GFR < 30 mL/min	527	843	<0.0001

Abbreviations. GFR = glomerular filtration rate, LV-EF = left ventricular ejection fraction, NSTEMI = non-ST-elevation myocardial infarction, and STEMI = ST-elevation myocardial infarction.

**Table 3 jcm-12-00917-t003:** Survival of patients regarding the presence or absence of diabetes mellitus in patients following different kinds of myocardial infarction, in patients with reduced vs. normal left ventricular function and in patients with reduced vs. normal renal function.

	n	Mean FU (Days)	Minimum FU (Days)	Maximum FU (Days)	SD
STEMI with DM	2244	1770.6	1.0	4332.2	1265.5
STEMI without DM	2943	2206.3	1.0	4342.0	1198.1
NSTEMI with DM	1901	1341.4	1.0	4299.8	1077.5
NSTEMI without DM	3688	1772.6	1.0	4325.1	1126.6
DM with reduced LVF	956	970.6	1.0	4262.1	1061.0
DM with normal LVF	2866	1784.5	1.0	4332.2	1168.5
no DM with reduced LVF	850	1203.6	1.1	4244.1	1106.7
no DM with normal LVF	5152	2085.0	1.0	4342.0	1121.6
DM with GFR > 60 mL/min	2830	1863.5	1.0	4332.2	1146.3
DM with GFR 30–60 mL/min	1245	1080.5	1.0	4305.4	1118.2
DM with GFR < 30 mL/min	253	526.8	1.0	4299.7	776.5
no DM with GFR > 60 mL/min	5549	2152.6	1.0	4342.0	1127.5
no DM with GFR 30–60 mL/min	1158	1351.5	1.0	4325.0	1124.5
no DM with GFR < 30 mL/min	254	843.6	1.0	4162.6	983.3

Abbreviations. DM = diabetes mellitus, GFR = glomerular filtration rate, FU = follow-up, LVF = left ventricular function, NSTEMI = non-ST-elevation myocardial infarction, STEMI = ST-elevation myocardial infarction.

## Data Availability

Not appilicable.

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
