# Peer review of "Survival of Myocardial Infarction Patients with Diabetes Mellitus at the Invasive Era (Results from the Városmajor Myocardial Infarction Registry)"

_jcm, 2023, doi:10.3390/jcm12030917_

Round 1

Reviewer 1 Report (Previous Reviewer 2)

The authors should re-check the manuscript for typing errors, in the references section and title. Authors must use the whole word in the title. The use of the abbreviation is very difficult and ambiguous for the reader in the title. For example, please use the whole word VMAJOR-MI, MI and DM in the title and running title.

Please use the correct name of the author in reference lists 2, 5 and 11. For example, you correct the first name and last name of   M.H. Seyed Mohammadzad in reference numbers 14 and 15. Also, you must correct the author’s name in all reference lists.

Please add this reference that is supportive and applicable to your paper: https://doi.org/10.1016/j.dsx.2022.102602.

Author Response

The authors should re-check the manuscript for typing errors, in the references section and title. Authors must use the whole word in the title. The use of the abbreviation is very difficult and ambiguous for the reader in the title. For example, please use the whole word VMAJOR-MI, MI and DM in the title and running title.

Thanks for comments. All corrections have been performed.

Please use the correct name of the author in reference lists 2, 5 and 11. For example, you correct the first name and last name of   M.H. Seyed Mohammadzad in reference numbers 14 and 15. Also, you must correct the author’s name in all reference lists.

The name of authors has been corrected in all references.

Please add this reference that is supportive and applicable to your paper: https://doi.org/10.1016/j.dsx.2022.102602.

Thanks for this comment, it has been referred as suggested.

Mansouri F, Mohammadzad MHS. Effects of metformin on changes of miR-19a and miR-221 expression associated with myocardial infarction in patients with type 2 diabetes. Diabetes Metab Syndr 2022; 16: 102602.

Reviewer 2 Report (Previous Reviewer 1)

This is my second review of this manuscript – I wrote an earlier one before returning it. This one should be analyzed together with the previous one (under the same number).

I stand by my earlier review - the paper presents an important analysis on the impact of DM on the survival of post-MI patients treated with PCI. Clinical factors such as MI type, renal function and LVEF were considered. Noteworthy is the large population size.

The authors have taken into account my previous comments and made appropriate revisions to the manuscript, which improve its scientific merit. Nevertheless, the discussion could still be broadened and expanded to include aspects of the impact of DM on post-MI survival (e.g., discuss STEMI and NSTEMI groups).

I still find the paper valuable.

Author Response

This is my second review of this manuscript – I wrote an earlier one before returning it. This one should be analyzed together with the previous one (under the same number).

I stand by my earlier review - the paper presents an important analysis on the impact of DM on the survival of post-MI patients treated with PCI. Clinical factors such as MI type, renal function and LVEF were considered. Noteworthy is the large population size.

The authors have taken into account my previous comments and made appropriate revisions to the manuscript, which improve its scientific merit. Nevertheless, the discussion could still be broadened and expanded to include aspects of the impact of DM on post-MI survival (e.g., discuss STEMI and NSTEMI groups).

 I still find the paper valuable.

Thanks for the nice words. Discussion section  has been extended as required.

Reviewer 3 Report (New Reviewer)

In this observational study, Skoda et al. have shown that diabetes mellitus (DM) imposes a strong negative prognostic impact on myocardial infarction (MI) subtypes who underwent percutaneous coronary intervention (PCI). In addition, the left ventricular and renal functions are shown to critical for the patients’ survival over the presence of DM. Overall, this study seems very robust, and is critical for an improved medical intervention in the patients with DM and MI.

Major:

      I.         Mechanistically, it is not clear why non-ST-elevation MI (NSTEMI) shows worse 1-year prognosis than STEMI in both the diabetic and non-diabetic subjects. The authors have mentioned that NSTEMI patients show more vascular complications compared to the STEMI. However, it is worth to briefly discuss how dysfunctional vasculature, in T2DM or not, can potentially exacerbate the pathology of NSTEMI.

Minor:

      I.         It is not clear whether all the patients in this study had type 2 DM or not? It would be better to specify T2DM instead of DM if such is the case.

     II.         Page 2: Missing “,” “Due to a lifelong nature of diabetes mellitus (DM) it has been” and “A total of 12 270 patients”

   III.         Page 8: “Later” not “latter”

   IV.         Circulating MicroRNAs are known as the early diagnosis and prognosis markers of acute MI. It would be a helpful future study to track the expressions of microRNA-1 and microRNA-221-3p in the PCI-MI patients with or without DM to avoid the confounding effect.

Author Response

In this observational study, Skoda et al. have shown that diabetes mellitus (DM) imposes a strong negative prognostic impact on myocardial infarction (MI) subtypes who underwent percutaneous coronary intervention (PCI). In addition, the left ventricular and renal functions are shown to critical for the patients’ survival over the presence of DM. Overall, this study seems very robust, and is critical for an improved medical intervention in the patients with DM and MI.

Thanks for the nice words.

Major:

  1. Mechanistically, it is not clear why non-ST-elevation MI (NSTEMI) shows worse 1-year prognosis than STEMI in both the diabetic and non-diabetic subjects. The authors have mentioned that NSTEMI patients show more vascular complications compared to the STEMI. However, it is worth to briefly discuss how dysfunctional vasculature, in T2DM or not, can potentially exacerbate the pathology of NSTEMI.

Thanks for this comment. The text has been extended with the followings as suggested:

There is a substantial difference in pathomechanism between STEMI and NSTEMI with less total acute occlusion and more extensive vascular disease with multiple narrowing in the later (17,18). Advanced DM accelerates vascular smooth muscle cell dysfunction, which contributes to the development of vasculopathy (19).  Due to the effects of DM on vascular atherosclerosis, worse prognosis was expected and found in NSTEMI (20).

Minor:

  1. It is not clear whether all the patients in this study had type 2 DM or not? It would be better to specify T2DM instead of DM if such is the case.

Thanks for this comment. A total of 12,270 patients with STEMI or NSTEMI have been involved in this study, from which 4388 had DM, while 7018 cases had no DM (see Table 1). Unfortunately, there is no information on the type of DM in all cases, therefore the term ‘DM’ was used throughout the manuscript. These facts are now mentioned in the text:

In this pool of patients, 4388 subjects had DM, while 7018 cases had no DM.

In the Limitations:

Unfortunately, there was no information on the type of DM in all cases, therefore the term ‘DM’ was used throughout the manuscript.

  1. Page 2: Missing “,” “Due to a lifelong nature of diabetes mellitus (DM) it has been” and “A total of 12 270 patients”

Thanks for this comment. Corrections have been done as suggested.

  • Page 8: “Later” not “latter”

Thanks for this comment. Corrections have been done as suggested.

  1. Circulating MicroRNAs are known as the early diagnosis and prognosis markers of acute MI. It would be a helpful future study to track the expressions of microRNA-1 and microRNA-221-3p in the PCI-MI patients with or without DM to avoid the confounding effect.

Thanks for this comment.  We absolutely agree with the reviewer’s suggestion. This sort of study would help understanding the role of microRNAs in such subgroup of MI patients. It could be a topic of future investigations.

Reviewer 4 Report (New Reviewer)

In the manuscript "Survival of myocardial infarction patients with diabetes mellitus at the invasive era" the Authors have addressed about the impact of DM on clinical outcomes of MI patients treated with PCI. The significance of this study is to analyze the association between two clinically important diseases. However, prior to the publication, substantial changes should be introduced into the manuscript.

1.     Since there have been several studies regarding diabetes and myocardial infarction, the authors need to clarify the novelty of this study. In addition, the authors presented only relatively short 1-year f/u data based on relatively old registry data with significant differences compared to current PCI technique, device and drug treatment.

2.     No baseline data were presented for each patient group.

3.     The authors described ‘the presence of DM seems to be more important than the type of MI’ in terms of survival based on Kaplan-Meier curves. However, the KM curve alone is not appropriate for drawing such a conclusion. It is thought that other statistical methods are needed. (Following paper might be helpful for authors: “Comparison of Clinical Outcomes after Non-ST-Segment and ST-Segment Elevation Myocardial Infarction in Diabetic and Nondiabetic Populations”, J Clin Med. 2022 Aug 29;11(17):5079. doi: 10.3390/jcm11175079.)

4.     For the same reasons as above, evidences for the following conclusions are insufficient.

- ‘Regarding to the survival analysis, it is primarily affected by LV function, than DM’
- ‘Regarding to survival analysis, it is primarily affected by renal function, than DM’

5.     In the discussion section, it is necessary to analyze the results and present their meaning rather than repeating the data already presented by the authors in the text. In particular, the part about new diabetes drugs or new MI diagnostic techniques lacks logical connection with the contents of the manuscript

6.     This study does not provide any basis for the authors’ conclusions as follows;
“These results could highlight our attention on the importance of recent DM treatment with new drugs including SGLT-2 inhibitors and GLP-1 antagonists with beneficial effects on survival.”

Author Response

In the manuscript "Survival of myocardial infarction patients with diabetes mellitus at the invasive era" the Authors have addressed about the impact of DM on clinical outcomes of MI patients treated with PCI. The significance of this study is to analyze the association between two clinically important diseases. However, prior to the publication, substantial changes should be introduced into the manuscript.

  1. Since there have been several studies regarding diabetes and myocardial infarction, the authors need to clarify the novelty of this study. In addition, the authors presented only relatively short 1-year f/u data based on relatively old registry data with significant differences compared to current PCI technique, device and drug treatment.

Thanks for this comment. The novelty of the present study is mentioned in the Conclusion section emphasizing that DM is a more important factor than the type of MI for prognosis and survival is primarily affected by LV and renal functions, than DM. These are new findings, which seem to be important to publish as confirmed by other reviewers. On the other side, 1-year follow-up is a relatively long follow-up in a population of more than 12,000 MI patients from the largest single-center in Hungary. It should also be considered, that due to pandemic, such studies could not be organized in the last years due to other important clinical work of clinical scientists.

  1. No baseline data were presented for each patient group.

Thanks for this comment. The most important baseline data including mean age, gender, presence of type of MI, ventricular fibrillation, cardiogenic shock and heart failure and LVEF are presented in Table 1.

  1. The authors described ‘the presence of DM seems to be more important than the type of MI’ in terms of survival based on Kaplan-Meier curves. However, the KM curve alone is not appropriate for drawing such a conclusion. It is thought that other statistical methods are needed. (Following paper might be helpful for authors: “Comparison of Clinical Outcomes after Non-ST-Segment and ST-Segment Elevation Myocardial Infarction in Diabetic and Nondiabetic Populations”, J Clin Med. 2022 Aug 29;11(17):5079. doi: 10.3390/jcm11175079.)

Thanks for this comment. When the statistical analysis was done, we followed statisticians’ suggestions, therefore Kaplan-Meier curves were generated during evaluations according to its definitions: The Kaplan–Meier estimator, also known as the product limit estimator, is a non-parametric statistic used to estimate the survival function from lifetime data. We think it is important to follow suggestions of an expert in this field. The cited paper examined similar problems but from different aspects of view, but we do not want to duplicate these results from another working group in a similar topic.

  1. For the same reasons as above, evidences for the following conclusions are insufficient.

    - ‘Regarding to the survival analysis, it is primarily affected by LV function, than DM’
    - ‘Regarding to survival analysis, it is primarily affected by renal function, than DM’

Thanks for this comment. However, we do not agree with reviewers’ opinions. Other reviewers totally agreed with our findings and conclusions.

  1. In the discussion section, it is necessary to analyze the results and present their meaning rather than repeating the data already presented by the authors in the text. In particular, the part about new diabetes drugs or new MI diagnostic techniques lacks logical connection with the contents of the manuscript
  2. This study does not provide any basis for the authors’ conclusions as follows;
    “These results could highlight our attention on the importance of recent DM treatment with new drugs including SGLT-2 inhibitors and GLP-1 antagonists with beneficial effects on survival.”

Thanks for this comment. However, these parts of manuscript have been inserted into the text according to other reviewers’ suggestions, who found this sort of extension important to understand results.

Round 2

Reviewer 4 Report (New Reviewer)

The authors provided their opinions for the each review comments.

I respect the efforts of the authors.

This manuscript is a resubmission of an earlier submission. The following is a list of the peer review reports and author responses from that submission.

Round 1

Reviewer 1 Report

This paper presents relevant results for the study population on the impact of the presence of DM on survival of patients after MI treated with PCI, taking into account factors such as type of MI, LVEF and renal function. The study population is large (n= 12,270) and representative of both MI types: NSTEMI (n=6840) and STEMI (n=5430). However, a considerable limitation of the study (pointed out by the authors themselves) is that all causes of death were included, not just cardiovascular deaths.

The conclusions presented are relevant to the results obtained, which do not raise any doubt. 

However, I do have a few comments on the reviewed manuscript:

1. There are no references to Tables 3, 4 and 5 in the relevant text. These tables could in principle be combined into one (with proper editing).

2. The figures included in the review file are of very poor technical quality, making them difficult to read.

3. In the proper text, the methodological description lacks, in my opinion, an indication of where the authors obtained their knowledge of the deaths of the subjects - given the purpose of the study, such information is important. 

4. In my opinion, the discussion makes very limited reference to the results obtained and disproportionately describes the issues related to the applicability of SGLT-2 inhibitors and GLP-1 antagonists. With a view to analysing all deaths, not just cardiovascular deaths, the authors should have focused more on that issue. 

Despite the comments indicated, I consider the work to be valuable, but in need of some adjustments.

Author Response

This paper presents relevant results for the study population on the impact of the presence of DM on survival of patients after MI treated with PCI, taking into account factors such as type of MI, LVEF and renal function. The study population is large (n= 12,270) and representative of both MI types: NSTEMI (n=6840) and STEMI (n=5430). However, a considerable limitation of the study (pointed out by the authors themselves) is that all causes of death were included, not just cardiovascular deaths.

The conclusions presented are relevant to the results obtained, which do not raise any doubt.

However, I do have a few comments on the reviewed manuscript:

1. There are no references to Tables 3, 4 and 5 in the relevant text. These tables could in principle be combined into one (with proper editing).

We have corrected it in the text, we think that the table should be separate due to good transparency.

2. The figures included in the review file are of very poor technical quality, making them difficult to read.

We have re-inserted the figures

3. In the proper text, the methodological description lacks, in my opinion, an indication of where the authors obtained their knowledge of the deaths of the subjects - given the purpose of the study, such information is important.

We refer to it in the Methods section

4. In my opinion, the discussion makes very limited reference to the results obtained and disproportionately describes the issues related to the applicability of SGLT-2 inhibitors and GLP-1 antagonists. With a view to analysing all deaths, not just cardiovascular deaths, the authors should have focused more on that issue.

During the examined period, the above-mentioned medicines were not yet available in Hungary

Despite the comments indicated, I consider the work to be valuable, but in need of some adjustments.

Reviewer 2 Report

I regret the manuscript titled:

 Survival of myocardial infarction patients with diabetes mellitus at the invasive era, has a good quality for publication in your journal.

It needs to:

1)   The authors should re-check the manuscript for typing errors, scientific English and grammar in the whole paper. For example, in the abstract this sentence has a typing mistake ‘treatment with new drugs insluding SGLT-2 inhibitors” and other places.

2)   The authors must change the running title.

3) The authors should explain about MicroRNA in the main text in a concise only one paragraph and you can use these references that are supportive and applicable in your paper: -Molecular miR-19a in Acute Myocardial Infarction: Novel Potential Indicators of Prognosis and Early Diagnosis. DOI:10.31557/APJCP.2020.21.4.975.

Up-Regulation of Cell-Free MicroRNA-1 and MicroRNA-221-3p Levels in Patients with Myocardial Infarction Undergoing Coronary Angiography. DOI: 10.34172/apb.2021.081.

I regret the manuscript titled:

 Survival of myocardial infarction patients with diabetes mellitus at the invasive era, has a good quality for publication in your journal.

It needs to:

1)   The authors should re-check the manuscript for typing errors, scientific English and grammar in the whole paper. For example, in the abstract this sentence has a typing mistake ‘treatment with new drugs insluding SGLT-2 inhibitors” and other places.

2)   The authors must change the running title.

3) The authors should explain about MicroRNA in the main text in a concise only one paragraph and you can use these references that are supportive and applicable in your paper: -Molecular miR-19a in Acute Myocardial Infarction: Novel Potential Indicators of Prognosis and Early Diagnosis. DOI:10.31557/APJCP.2020.21.4.975.

Up-Regulation of Cell-Free MicroRNA-1 and MicroRNA-221-3p Levels in Patients with Myocardial Infarction Undergoing Coronary Angiography. DOI: 10.34172/apb.2021.081.

Author Response

I regret the manuscript titled:

 Survival of myocardial infarction patients with diabetes mellitus at the invasive era, has a good quality for publication in your journal.

It needs to:

  • The authors should re-check the manuscript for typing errors, scientific English and grammar in the whole paper. For example, in the abstract this sentence has a typing mistake ‘treatment with new drugs insluding SGLT-2 inhibitors” and other places.

We corrected the typing errors.

  • The authors must change the running title.

Corrected

3) The authors should explain about MicroRNA in the main text in a concise only one paragraph and you can use these references that are supportive and applicable in your paper: -Molecular miR-19a in Acute Myocardial Infarction: Novel Potential Indicators of Prognosis and Early Diagnosis. DOI:10.31557/APJCP.2020.21.4.975.

Up-Regulation of Cell-Free MicroRNA-1 and MicroRNA-221-3p Levels in Patients with Myocardial Infarction Undergoing Coronary Angiography. DOI: 10.34172/apb.2021.081.

We were unable to carry out the following tests, so unfortunately we cannot refer to them in the article